# Fault Diagnosis of a Reciprocating Compressor Air Valve Based on Deep Learning

**Shungen Xiao** [1,2,3], **Ang Nie** [2], **Zexiong Zhang** [1,3] , **Shulin Liu** [2,*], **Mengmeng Song** [1,*] **and Hongli Zhang** [2,*]

[1] College of Information, Mechanical and Electrical Engineering, Ningde Normal University, Ningde 352100, Fujian, China; xiaoshungen022@163.com (S.X.); 3191218017@fafu.edu.cn (Z.Z.)
[2] School of Mechatronic Engineering and Automation, Shanghai University, Shanghai 200072, China; nieang@csvw.com
[3] College of Mechanical and Electrical Engineering, Fujian Agriculture and Forestry University, Fuzhou 350000, Fujian, China
[*] Correspondence: lsl346@shu.edu.cn (S.L.); songmengmeng_022@163.com (M.S.); meiliwenhui@shu.edu.cn (H.Z.)

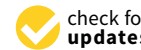

**Featured Application: The method proposed in the article can judge the fault type of the reciprocating compressor without damage through the collected electrical signals.**

**Abstract:** With the development of machine learning in recent years, the application of machine learning to machine fault diagnosis has become increasingly popular. Applying traditional feature extraction methods for complex systems will weaken the characterization capacity of features, which are not conducive to subsequent classification work. A reciprocating compressor is a complex system. In order to improve the fault diagnosis accuracy of complex systems, this paper does not use traditional fault diagnosis methods and applies deep convolutional neural networks (CNNs) to process this nonlinear and non-stationary fault signal. The valve fault data is obtained from the reciprocating compressor test bench of the Daqing Natural Gas Company. Firstly, the single-channel vibration signal is collected on the reciprocating compressor and the one-dimensional CNN (1-D CNN) is used for fault diagnosis and compared with the traditional model to verify the effectiveness of the 1-D CNN. Next, the collected eight channels signals (three channels of vibration signals, four channels of pressure signals, one channel key phase signal) are applied by 1-D CNN and 2-D CNN for fault diagnosis to verify the CNN that it is still suitable for multi-channel signal processing. Finally, further study on the influence of the input of different channel signal combinations on the model diagnosis accuracy is carried out. Experiments show that the seven-channel signal (three-channel vibration signal, four-channel pressure signal) with the key phase signal removed has the highest diagnostic accuracy in the 2-D CNN. Therefore, proper deletion of useless channels can not only speed up network operations but also improve diagnosis accuracy.

**Keywords:** reciprocating compressor; fault diagnosis; deep learning; CNN

## 1. Introduction

A reciprocating compressor is the most widely used compressor type in industry and key equipment in gas transmission pipelines, petrochemical industry, fertilizer industry, oil refinery, ethylene chemical industry, coal chemical industry, and other industries. Monitoring and fault diagnosis of reciprocating compressors can help the machine to continue normal operation and it is of great significance. For reciprocating compressors, the gas valve is one of the components with the highest failure rate in the reciprocating compressor in [1].

Research on the fault diagnosis of the reciprocating compressor valve mainly focuses on three aspects: Vibration monitoring, thermal performance monitoring, and the indicator diagram.

(a) Vibration monitoring method: Gas valve faults often occur together with abnormal vibration signal. Therefore, using vibration signal for analysis is the most common method for diagnosing gas valve faults. Jiang et al. [2] used basis pursuit to extract the principal components of the vibration signal and employed waveform matching to extract the signal features. Finally, he applied support vector machines to identify the valve failure mode. Kurt et al. [3] transformed the vibration signal of the valve into a high-dimensional vector space and defined the metric in this space. Finally, the distance between the actual state of the compressor and the reference state was calculated to determine whether a failure had occurred. Shao et al. [4] used the vibration signal of the valve of the reciprocating compressor to the experiment. Firstly, he performed wavelet packet decomposition to extract features. Next, he applied principal component analysis to reduce the dimensionality of the obtained feature and then input the BP neural network for diagnosis. The weight optimization of the network used a combination of particle swarm optimization and the genetic algorithm, and the classification accuracy reached 100%. Cerrada et al. [5] employed the symbolic dynamics and complex correlation measure to extract the features of vibration data as input to two random forest algorithms (ensemble subspace k nearest neighbor and ensemble bagged tree). The experimental results achieved a greater than 93% classification accuracy.

(b) Thermal performance method: The thermal performance parameters of reciprocating compressors mainly include the component temperature, lubricating oil temperature, exhaust volume during operation, exhaust pressure, cylinder pressure, etc. Gord [6] established a zero-dimensional numerical model of a single-stage reciprocating natural gas compressor. Next, he set a hole on the valve plate to simulate valve leakage and monitored the changes in temperature, pressure, and mass flow parameters at the inlet and outlet of the valve to diagnose valve faults. The model was verified by comparison with the existing experimental data. Finally, he proposed that the valve could be diagnosed by detecting the temperature change of the valve to diagnose whether the valve was faulty. Wang [7] established a mathematical model of the working cycle of a reciprocating compressor and applied software to simulate the dynamic pressure curve of the compressor under the condition of air volume adjustment. Then, he added the fault influences parameters to improve the mathematical model. This simulated the leakage fault of the suction valve and the spring stiffness failure and the cylinder pressure data under failure was obtained. Finally, based on the principal component analysis method, he classified the valve fault status. However, if the method based on the above parameters is used in the early stage of the failure, these thermal performance parameters do not change significantly, and the failure cannot be predicted well. Therefore, professional technicians are required to monitor relevant parameters and it is difficult to achieve real-time diagnosis. Besides, measuring parameters require many sensors and the cost is relatively high.

(c) Indicator diagram: The indicator diagram reflects the change curve of the piston position and the corresponding pressure in the cylinder in a working cycle of the reciprocating compressor. The piston position can be expressed by the cylinder gas volume, crank angle, piston stroke, etc. For reciprocating compressors, the thermal performance can reflect various types of faults and the indicator diagram can reflect many changes in thermal performance. Therefore, once the gas valve fails, the shape of the indicator diagram will change accordingly. This is the reason why many scholars have used indicator diagrams to perform fault diagnosis as well as carrying out a lot of research. Tang et al. [8] used backpropagation neural network(BPNN) to identify the fault type based on extracting the features of the geometric properties of the indicator diagram. He effectively diagnosed four common faults of the gas valve. Feng et al. [9] proposed curvelet transform to extract the typical features in the indicator diagram and reduced the dimensions of the high-dimensional features through principal component analysis. Next, he input features to support vector machines (SVM) for failure pattern recognition. This was

effective in discriminating clogged exhaust valves, clogged intake valves, and leaked exhaust valves, etc. However, the indicator diagram converts 1-D signal into 2-D signal, which increases the complexity of feature extraction. The extracted features are difficult to fully cover all the information of the indicator diagram. The information on the pressure in the cylinder and the position of the piston is difficult to collect. As a result, diagnosis by the indicator diagram has great limitations.

Xiao et al. [10] established a dynamic model of coupling translation joints with subsidence for time-varying load in a planar mechanical system. He discussed 42 kinds of coupling rub–impact scenarios of double translational joints with subsidence. Finally, Xiao et al. [11] employed the Poincaré cross-section method and the maximum Lyapunov exponent method to prove the chaotic behavior of the reciprocating compressor system. Wei et al. [12] explored the causes of self-excited oscillation caused by the bearing by establishing a mathematical model of the rotor system and proves the instability of the system. The chaotic behavior and the instability of system is often one of the reasons that make the signal more complex. Therefore, the fault signal of a reciprocating compressor has complex nonlinear and non-stationary characteristics. In addition, the signal contains a lot of noise, and some traditional signal processing methods are not applicable. For instance, fast Fourier transform (FFT) can be effectively applied to rotating machinery, but it cannot achieve good results when processing reciprocating compressor signals. Hence, alternative methods are used for processing nonlinear or transient signals, such as wavelet transform (WT) and empirical mode decomposition (EMD). They decompose the signal into different frequency bands and extract steady-state and linear features from different frequency bands. Jin et al. [13] used the wavelet transform with the basis function of Bior3.5 wavelet to decompose the original vibration signal into various frequency bands. Then, he utilized energy spectrum analysis and used the energy spectrum features as the input of the support vector machine for diagnosis. The experimental results show that the accuracy of fault identification exceeds 90%. Lin [14] employed ensemble empirical mode decomposition (EEMD) to decompose the vibration signal into intrinsic modal functions (IMFs) of different frequencies and used the Hilbert spectrum to extract the fault characteristics of the natural gas compressor. However, it is difficult to select the basis function of the wavelet transforms and there is no standard to select the basis function for different signals. EMD also has many problems, such as modal aliasing. In order to deal with the transient characteristics of the signal, some scholars have proposed other methods to extract the faulty feature of the vibration signal of the reciprocating compressor. A feature extraction method of local maximum multi-scale entropy and extended multi-scale entropy was proposed by Zhao et al. [15]. The extracted features can characterize bearing faults well. Van et al. [16] proposed the Teager–Kaiser energy operator to simultaneously extract the vibration signal, pressure signal, and current signal of the reciprocating compressor and then used the deep belief network to identify the fault. Tang et al. [17] proposed adaptive peak decomposition to extract the characteristics of the vibration signal of the four-state reciprocating compressor. Qi et al. [18] sparsely encoded the reciprocating compressor operating data in 5 years and identified the faults using SVM. Tang et al. [19] calculated the normalized Lempel–Ziv complexity of the signal using the mean symbolization method and applied artificial neural networks to diagnose the unit fault.

The intelligent fault diagnosis of the reciprocating compressor includes two steps: Feature extraction and pattern recognition. Features are mainly extracted in the time domain, frequency domain, and time-frequency domain. The effectiveness of feature extraction directly affects the subsequent diagnosis results. Because the reciprocating compressor contains many rotating and moving parts, it would cause many vibration sources and the measured vibration signal usually includes transient vibration and noise. As a result, the frequency bandwidth of the signal has a complicated shape as well as showing nonlinear characteristics. Hence, conventional intelligent fault diagnosis methods are often not suitable for reciprocating compressors and affect the accuracy of failure mode identification.

In recent years, as the concept of deep learning has been proposed, it has been widely used in various fields because deep learning can effectively self-extract features and classify data. It is also widely used in machine fault diagnosis. In some more complex mechanical systems, it can extract the good characteristics of system faults and then perform fault classification or predict the remaining life of the machine through regression. Guo [20] used a deep neural network model with an adaptive learning ability and hierarchical learning rate, of which the testing accuracy was over 99.3% in the bearing data set. Long- and short-term memory (LSTM) is very suitable for processing time series. It is a kind of deep feedback neural network, which can effectively extract the characteristics of short and long periods of signal. Therefore, it has been applied to fault diagnosis by many scholars. Diego et al. [21] proposed LSTM based on deep learning, which directly extracts the original vibration signal features of the reciprocating compressor and performs fault pattern recognition. He used Bayesian optimization to select the hyperparameters of the model and compared with it several machine learning methods. The comparison showed that his method obtained better diagnostic results. Shen [22] proposed a network called SeriesNet, which is specially used to process time series. It consists of an LSTM network and dilated causal convolutional neural network. It effectively predicts several stock and regional temperature data sets and predicts the results. It is more consistent than the typical ANN, SVM, and other prediction models. The convolutional neural network (CNN) is different from LSTM. It is a kind of feedforward neural network. It pays more attention to the local characteristics of data, and the effectiveness of the CNN has been proved in the field of fault diagnosis. Long et al. [23] applied a CNN with a LeNet-5 architecture to convert the original vibration signal into an image. Furthermore, he used this model for feature self-extraction and pattern recognition. As a result, he achieved high recognition accuracy in the data sets of three different machines.

In most cases, a signal cannot reflect a potential failure because the failure of a reciprocating compressor is often caused by the interaction of different factors and parameters. The collected signal also usually contains information parameters, such as vibration, temperature, and pressure. The relationship between these parameters is very complex and affects each other. Hence, it is necessary to perform comprehensive fault recognition on multi-source signal data. In this paper, the CNN on deep learning is utilized to realize the fault diagnosis of the air valve of the reciprocating compressor.

In order to solve the above problems, this article mainly studies from the following aspects:

(1) A 1-D CNN model is built and the original vibration signal of a single measuring point (only one sensor is used to collect a one channel signal) is employed to train the network. Next, a nonlinear mapping of the original vibration signal to the fault type is established. CNN is applied to extract features of the original signal and realize fault diagnosis of the reciprocating compressor gas valve. The experiment was conducted on the condition of a single measuring point.

(2) Eight-channel signals using multiple measuring points (four vibration sensors, four pressure sensors, one key phase sensor) are applied in a 1-D CNN model and a 2-D CNN model for reciprocating compressor valve fault diagnosis. Furthermore, this article studies the influence of different fusion measuring points signals.

## 2. Convolutional Neural Network

CNN is a neural network specially used to process data with similar grid structures, for example, 1-D time series data (which can be considered as a 1-D grid formed by regularly sampling on the time axis) and 2-D image data (which can be regarded as a 2-D pixel grid) [24]. CNN is a multi-layer neural network and it contains an input layer, hidden layers, and an output layer. The hidden layer is mainly composed of a convolution layer and a pooling layer. The hidden layer processes the input signal and extracts features. Finally, the fully connected layer acts as a classifier to realize the mapping with the output target.

### 2.1. Convolution Layer

The convolution operation expression for discrete data is given as follows:

$$s(t) = (x \times w)(t) = \sum_{a=-\infty}^{\infty} x(a) \times w(t-a) \tag{1}$$

where $x$ is the input and $w$ is the kernel function. Here, the $s$ output is called the feature map. Equation (1) is a convolution operation for 1-D data. When the input data are a multi-dimensional array, the parameters of the kernel are usually a multi-dimensional array (features) and trained by the learning algorithm. For example, the formula for 2-D convolution with a 2-D convolution kernel of a 2D image is as follows:

$$S(i,j) = (I \times K)(i,j) = \sum_{m}\sum_{n} I(i+m, j+n)K(m,n) \tag{2}$$

where $S(i, j)$ is the pixel value in the output image $S$ at the point $(i, j)$, and $K(m, n)$ is the parameter value of the convolution kernel at the point. Figure 1 shows an example of the convolution operation on a 2-D tensor.

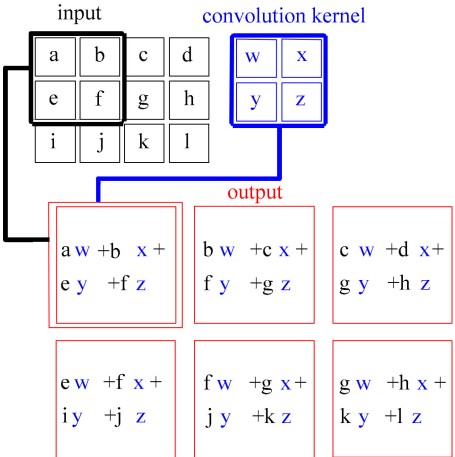

**Figure 1.** Schematic of the 2-D convolution operation.

The convolution operation makes the CNN have sparse interactions, parameter sharing, and equivariant presentations, etc. These properties make the convolution operation very suitable for processing time series data collected by reciprocating compressors.

Each convolutional layer may contain more than one convolution kernel and each convolution kernel corresponds to a channel of output data. The output after the convolution operation is necessary to add a bias term and is then activated by a nonlinear activation function. The output of the convolution layer is described as the following:

$$z_i^{[l]} = K_i^{[l]} \times a^{[l-1]} + b_i^{[l]} \tag{3}$$

$$a_i^{[l]} = g(z_i^{[l]}) \tag{4}$$

where $K_i^{[l]}$ is the $i$-th convolution kernel in layer $l$, and $a^{[l-1]}$ is the output of layer $l{-}1$. The offset term corresponding to the $i$-th convolution kernel in layer $l$ of $b_i^{[l]}$. $z_i^{[l]}$ is not the activated output of the channel $i$ in layer $l$. $g$ is the nonlinear activation function. $a^{[l]}$ is the output of the channel $i$ of the $l$-th layer and it is the input data of the next layer.

Among them, the activation function enables the model to learn the nonlinear mapping of the input data, which improves the model's learning features ability. Activation functions mainly include the sigmoid function, tanh function, linear rectification function (ReLU), leakage rectification

function, parametritis modified linear unit [25], softmax function, etc. Because ReLU does not easily disappear when using the error back propagation algorithm for model training, it can make shallow parameters easier to learn. Thus, the process of model training is accelerated, and the model converges better. Therefore, ReLU is also employed as the activation function in the convolutional layer and the expression of ReLU is as follows:

$$g(z) = \max(0, z) \tag{5}$$

For the convolutional layer, the parameters of each convolution kernel need to be obtained through training.

### 2.2. Pooling Layer

The pooling layer compresses the output of the previous layer. Concretely, it takes the overall statistics of the adjacent output of the convolutional layer at a certain position as the output of the pooling layer. The pooling layer has no parameters that need to be trained and pooling functions commonly include the maximum pooling function [26], average pooling function, norm, and weighted average function. Among them, the most commonly use pooling function is the max pooling, which uses a window to scan on the tensor and takes the maximum value in the window as the output. When the data passing through the pooling layer are 1-D, the max pooling function is as follows:

$$P_i^{[l]}(j) = \max_{(j-1)W+1 \leq t \leq jW} \left\{ a_i^{[l-1]}(t) \right\} \tag{6}$$

where $a_i^{[l-1]}(t)$ is output of the channel $i$ of the layer $[l-1]$ at point t and $W$ is the size of the pooled area. $P_i^{[l]}(j)$ is the output by the data at the point $j$ of the channel $i$ after passing through the pooling layer.

When the input data are slightly shifted or fluctuated, pooling can obtain an approximately constant output. Local translation invariance can make the model only care about whether a feature appears but not where it appears. This means that the use of the pooling layer can significantly compress data and reduce the size of the model. Hence, it can improve the calculation speed.

### 2.3. Fully Connected Layer

Features extracted through the pooling layer are input to the fully connected layer (dense layer) for compressing data and pattern classification. The fully connected layer is a multi-layer neural network and its structure is given in Figure 2. The number of layers and neurons in each layer are set manually and different layers as well as the number of neurons affect the accuracy of the network output. Each neuron is equivalent to an McCulloch-Pitts neuron Model (M-P Model), as shown in Figure 3 and receives input data transmitted from the previous layer of neurons. These data are transmitted through a connection with weight and the total input value received by the neuron needs to be added with a bias term and activated by a nonlinear activation function to generate the d into a column v output of the neuron. The M-P neuron model can be expressed as the following Equation (7):

$$y = g\left(\sum_{i=1}^{n} w_i a_i + b\right) \tag{7}$$

where $n$ is the number of neurons in the previous layer. $a_i$ is the output of the $i$-th neuron in the previous layer and $w_i$ is the weight corresponding to the $i$-th neuron in the upper layer. $b$ is the bias term. The calculation results inside the brackets get the output $y$ through the activation function $g$.

For each full connection layer, the weight parameter vector of Formula (7) is combined into a matrix $W^{[l]}$ and the bias $b$ is grouped into a column vector $b^{[l]}$. The output result after the activation function $g^{[l]}$ can be expressed as follows:

$$z^{[l]} = W^{[l]} a^{[l-1]} + b^{[l]} \tag{8}$$

$$a^{[l]} = g^{[l]}(z^{[l]}) \tag{9}$$

where the superscripts [*l*] and [*l* − 1] represent the layer where the parameters are located. $W^{[l]}$ is a matrix with dimensions (*n*, *m*). Among them, *n* and *m* are the number of neurons in the previous layer and the next layer, respectively. The output $a^{[l-1]}$ of the neuron in the previous layer undergoes a linear transformation of *W* and the bias vector $b^{[l]}$ to obtain an inactive output $z^{[l]}$. Then, the nonlinear mapping of $z^{[l]}$ through the activation function $g^{[l]}$ obtains the output $a^{[l]}$ of the current layer.

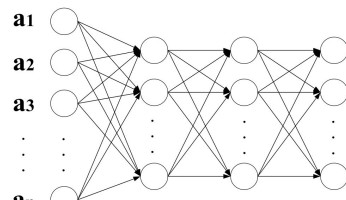

**Figure 2.** Fully connected layer structure.

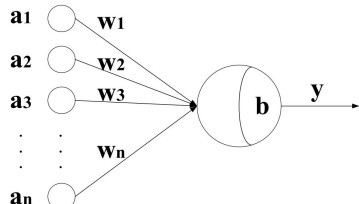

**Figure 3.** M-P neuron model.

The output of each neuron also needs to be activated by a nonlinear activation function that is like the convolutional layer. Because the different activation functions of each layer will affect the performance of the model, it is necessary to choose the appropriate activation function for different problems. In the output layer, if the study is preparing to solve the binary classification problem, the output layer generally contains only one neuron and the activation function is usually the sigmoid function as shown in Equation (10). However, the output layer generally contains multiple neurons and the activation function is usually the softmax function for multi-classification problems. The expression is given as Equation (11). Besides, the middle-hidden layer also usually selects ReLU as the activation function:

$$g(z) = \frac{1}{1 + e^{-z}} \tag{10}$$

$$g(z_i^{[l]}) = \frac{e^{z_i^{[l]}}}{\sum_{i=1}^{n^{[l]}} e^{z_i^{[l]}}} \tag{11}$$

where $z_i^{[l]}$ represents the output value of the *i*-th neuron in layer *l* before activation. For the fully connected layer, the weight matrix and the bias term denote the parameters that need to be obtained through training.

## 3. 1-D CNN Single-Measurement Point Diagnosis Model

The vibration signal of the reciprocating compressor is complex and difficult to get the features that have good characterization capacity. In this paper, a 1D CNN single-measuring point diagnosis model is proposed and its structure is shown in Figure 4. The end-to-end model using the original vibration signal as an input to 1D CNN to perform feature self-extraction and fault diagnosis.

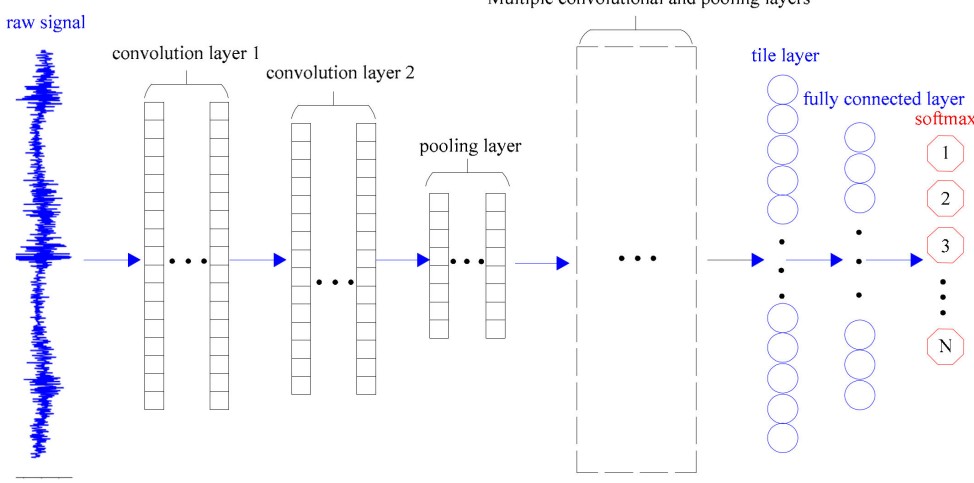

**Figure 4.** Schematic of a fault diagnosis model for a single measuring point reciprocating compressor in a 1D CNN.

The fault diagnosis model of 1D CNN single measuring point for reciprocating compressor valve includes 5 1D convolutional layers (Conv1D), 4 maximum pooling layers (MaxPooling 1D) and 2 fully connected layers (Dense). Among the model, the size of the convolutional layer and the convolution kernel both are 3 and the convolution step size is 1. Besides, the size of the pooling layer is 2 and the pooling step size is 2. Because this classification problem belongs to the 4 classifications problem, the output of the last layer is 4 neurons and uses softmax as the activation function. Therefore, the corresponding output is the probability value of each category and the category with the highest probability is the type of fault diagnosed. The specific parameters of the network are given in Table 1.

**Table 1.** 1D CNN single-point reciprocating compressor gas valve fault diagnosis model structure.

| Layer (Type) | Dimensions of Output Data | Number of Training Parameters |
|---|---|---|
| Input Layer | (None,1024, 1) | 0 |
| Conv1D 1 | (None, 1022, 32) | 128 |
| Conv1D 2 | (None, 1020, 64) | 6208 |
| MaxPooling1D 1 | (None, 510, 64) | 0 |
| Conv1D 3 | (None, 508, 128) | 24704 |
| MaxPooling1D 2 | (None, 254, 128) | 0 |
| Conv1D 4 | (None, 252, 128) | 49280 |
| MaxPooling1D 3 | (None, 126, 128) | 0 |
| Conv1D 5 | (None, 124, 64) | 24640 |
| MaxPooling1D 4 | (None, 62,64) | 0 |
| Tile Layer | (None, 3968) | 0 |
| Dense 1 | (None, 128) | 508032 |
| Dense 2 | (None, 4) | 516 |

In order to improve the generalization performance of the model, the model applies Dropout regularization, batch normalization and data augmentation technology. Data augmentation is to intercept part of the original signal by moving the interception window. When the moving window intercepts the original signal once, a new sample is obtained. Among them, 6964 samples were obtained after data augmentation and 70% of which were used as the training set and 30% as the test set.

## 4. 1-D CNN Multi-Measurement Point Diagnosis Model

Chen et al. [27] believe that due to the sensor layout and environmental interference, the collected vibration signals are different, which may lead to different diagnostic results. In order to improve the reliability of fault diagnosis, a new multi-sensor data fusion technology is proposed. Therefore, this

chapter changes the previous chapter's input signal to the signal collected by multiple sensors. Next, the collected signal is combined and input to the 1D CNN for fault diagnosis.

A schematic of a multi-point fault diagnosis model built using a 1-D CNN is shown in Figure 5. Here, the input data of the model is different from the previous section and it becomes an eight-channel signal. Therefore, the output value of each layer of neurons is the comprehensive response of the eight-channel signal. The convolution layer still uses a 1-D convolution kernel to perform the convolution operation.

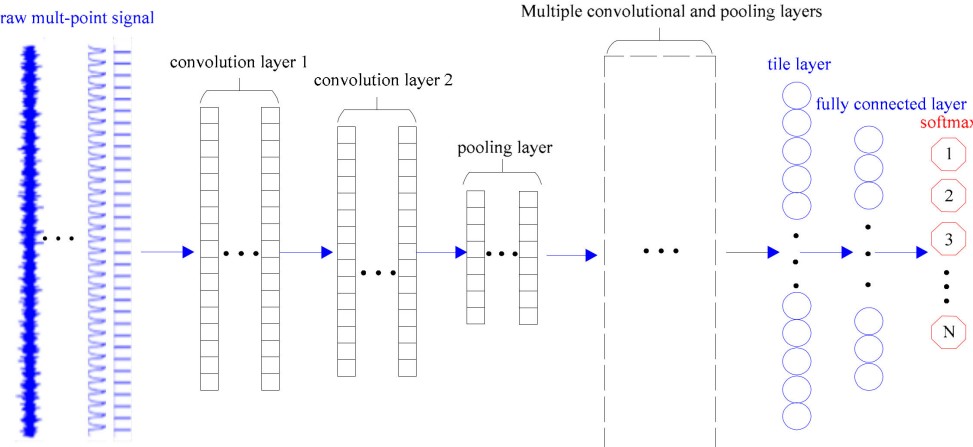

**Figure 5.** Reciprocating compressor valve fault diagnosis model of the 1-D CNN multi-measurement point.

If the input data of each channel of the model is still 1024 sampling points, the information of the pressure signal and the key phase signal cannot be fully expressed. Therefore, the input data of each channel of this model is increased to 3000 sampling points, making the dimension of the input data (3000, 8). Increased input data will cause the problem of too many parameters at the fully connected layer. In order to avoid this problem, the model adds the convolution layer and pooling layer to the model in the previous section to further compress the data features. The specific parameters of the model are shown in Table 2.

**Table 2.** 1-D CNN multi-measurement point reciprocating compressor gas valve fault diagnosis model structure.

| Layer (Type) | Dimensions of Output Data | Number of Training Parameters |
|---|---|---|
| Input Layer | (None, 3000, 8) | 0 |
| Conv1D 1 | (None, 2998, 32) | 800 |
| Batch Normalization 1 | (None, 2998, 32) | 128 |
| Conv1D 2 | (None, 2996, 64) | 6208 |
| Batch Normalization 2 | (None, 2996, 64) | 256 |
| MaxPooling1D 1 | (None, 1498, 64) | 0 |
| Conv1D 3 | (None, 1496, 128) | 24704 |
| Batch Normalization 3 | (None, 1496, 128) | 512 |
| MaxPooling1D 2 | (None, 748, 128) | 0 |
| Conv1D 4 | (None, 746, 128) | 49280 |
| Batch Normalization 4 | (None, 746, 128) | 512 |
| MaxPooling1D 3 | (None, 373, 128) | 0 |
| Conv1D 5 | (None, 371, 64) | 24540 |
| Batch Normalization 5 | (None, 371, 64) | 256 |
| MaxPooling1D 4 | (None, 185, 64) | 0 |
| Conv1D 6 | (None, 183, 64) | 12352 |
| Batch Normalization 6 | (None, 183, 64) | 256 |
| MaxPooling1D 5 | (None, 183, 64) | 0 |
| Tile layer | (None, 5824) | 0 |
| Dense 1 | None, 128) | 745600 |
| Dense 2 | (None, 4) | 516 |

## 5. Multi-Point Diagnosis Model of 2-D CNN

Different from the 1-D CNN in the previous chapter, the convolution kernel shape of 1-D CNN is a 2-D tensor $(m, n)$, where $n > 1$, and the convolution kernel used by 2-D CNN is a 3-D tensor $(m, n, l)$, where $n > 1$, $l \geq 1$. The number of channels output by each convolutional layer is the number of the the convolution kernel. Therefore, 1-D CNN and 2-D CNN have different numbers of channels in each convolutional layer.

The data of eight measuring points signal is combined into single-channel 2-D data (an image) and the 2-D convolution operation is used to process the image, as shown in Figure 6. The image data under four working conditions is the input of the model and the feature is extracted through 2-D CNN and then the failure mode is identified.

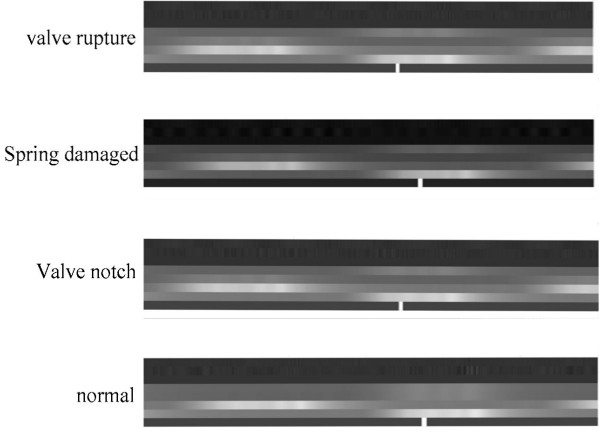

**Figure 6.** Model input data under four working conditions.

The schematic of a 2-D CNN multi-measuring point fault diagnosis model is shown in Figure 7. Because the input data is a one-channel gray scale image, the input dimension becomes (3000, 8, 1). Next, each convolutional layer performs a 2-D convolution operation and the input data is converted into multi-channel 2-D data by computing with different convolution kernels. Then, through the fully connected layer, the features extracted by the convolutional layer are further compressed and pattern recognition is performed. The specific parameters of the network are provided in Table 3 below.

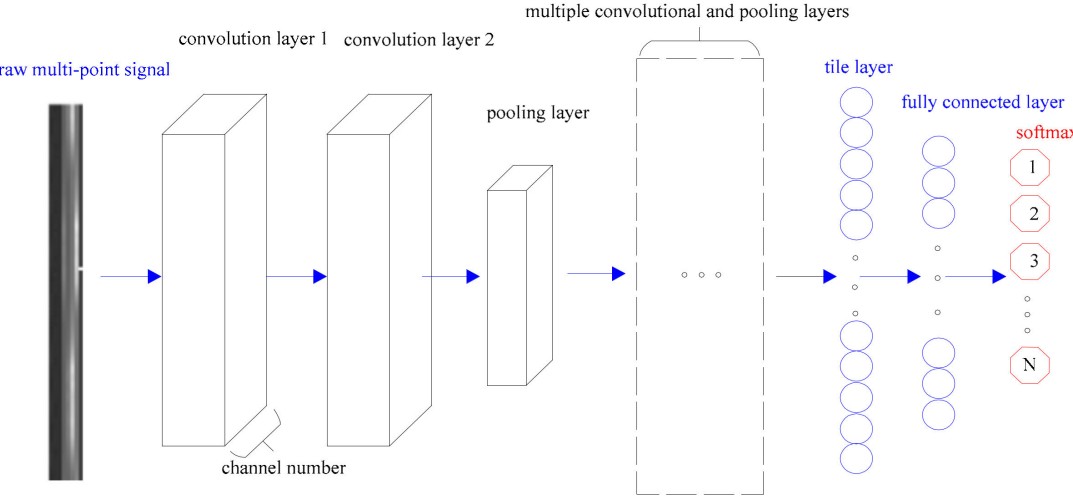

**Figure 7.** Reciprocating compressor valve fault diagnosis model of the 2-D CNN multi-measurement point.

**Table 3.** 2-D CNN multi-measurement point reciprocating compressor gas valve fault diagnosis model structure.

| Layer (Type) | Dimensions of Output Data | Number of Training Parameters |
|---|---|---|
| Input Layer | (None, 3000, 8,1) | 0 |
| Conv1D 1 | (None, 3000, 8,8) | 80 |
| Batch Normalization 1 | (None,3000, 8,8) | 32 |
| Conv1D 2 | (None, 3000, 8, 16) | 3216 |
| Batch Normalization 2 | (None, 3000, 8, 16) | 64 |
| MaxPooling1D 1 | (None, 1500, 8, 16) | 0 |
| Conv1D 3 | (None, 1500, 8, 32) | 4640 |
| Batch Normalization 3 | (None, 1500, 8, 32) | 128 |
| MaxPooling1D 2 | (None, 750, 8, 32) | 0 |
| Conv1D 4 | (None, 750, 8, 32) | 1056 |
| Batch Normalization 4 | (None, 750, 8, 32) | 128 |
| MaxPooling1D 3 | (None, 375, 8, 32) | 0 |
| Conv1D 5 | (None, 375, 8, 16) | 4624 |
| Batch Normalization 5 | (None, 375, 8, 16) | 64 |
| MaxPooling1D 4 | (None, 187, 8, 16) | (None, 187, 8, 8) |
| Conv1D 6 | (None, 187, 8, 8) | 1160 |
| Batch Normalization 6 | (None, 187, 8, 8) | 32 |
| MaxPooling1D 5 | (None, 93, 8, 8) | (None, 5952) |
| Tile layer | (None, 5952) | 0 |
| Dense 1 | (None, 128) | 761984 |
| Dense 2 | (None, 4) | 0 |

It can be seen from Table 3 that the model includes six 2-D convolutional layers (Conv2D), five maximum pooling layers (MaxPooling2D), and two fully connected layers. In order to reduce the amounts of parameters, the dimension of the convolution kernel is set to (3,3) and the convolution step size is 1. Next, the size of the pooling area of the pooling layer is designed as (2,1) and the pooling step is 2. The fully connected layer of the last layer uses softmax as the activation function, which is like the previous 1-D CNN model, but the remaining layers all use the ReLU function.

Because the input image of this model is oblong and the image will be compressed after each convolution, the key features at the edge of the image may be lost during convolution. In order to prevent this problem and make better use of edge data, the input of each convolutional layer is filled [28], which is to fill 0 around the image, as shown in Figure 8.

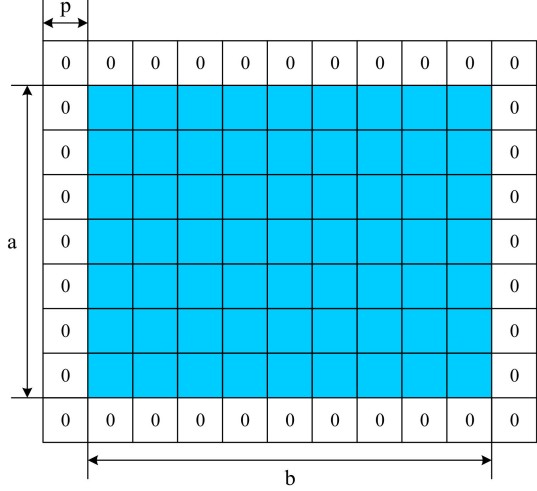

**Figure 8.** Padding diagrammatic sketch.

An image is convolved with a convolution kernel of size $(a, b)$. When the fill amount is $p$, the output image size is $(a + 2p − f + 1, b + 2p − f + 1)$. Here, the expectation is that the size of the image remains the same after convolution, so the parameters $p = 1$ and $f = 3$ are chosen to meet the requirements.

Overfitting is one of the common problems of deep learning models. Common solutions include increasing the amount of data or adding regularization methods. Therefore, dropout regularization, batch normalization, and data augmentation technology are applied to improve the generalization performance of the model, which is like a 1-D CNN measuring-points experiment. After using data augmentation technology, the data are sampled and 6964 samples are obtained. Among the samples, 70% of them are selected randomly as the training set and the remaining 30% of them as a test set.

## 6. Experimental Results and Discussion

### 6.1. Experimental Data Collection

It can be recognized from the most common failure type of the air valve of the reciprocating compressor that the damage of the valve plate and the spring is the most easily damaged part of the air valve. Therefore, fault simulation experiments were carried out on the secondary cylinder for the four types of normal valve, broken valve, spring damage, and notched valve.

This experiment was carried out in the working unit of No. 1 compressor station of the South District of Daqing Natural Gas Company in China and the basic structure as well as the sensor (measuring points) arrangement of the reciprocating compressor is shown in Figure 9. These sensors were used to collect vibration signals, pressure signals, and key phase signals. Several acceleration sensors were utilized to collect vibration signals. The acceleration sensor and the pressure sensor are installed on the pressure valve, and the key phase sensor is installed near the shaft between the motor and the crankcase. The specific positions of the valves and sensors on the primary and secondary cylinders are as depicted in Figure 9. In addition, the actual positions of the valves and sensors on the secondary cylinder are shown in Figure 10.

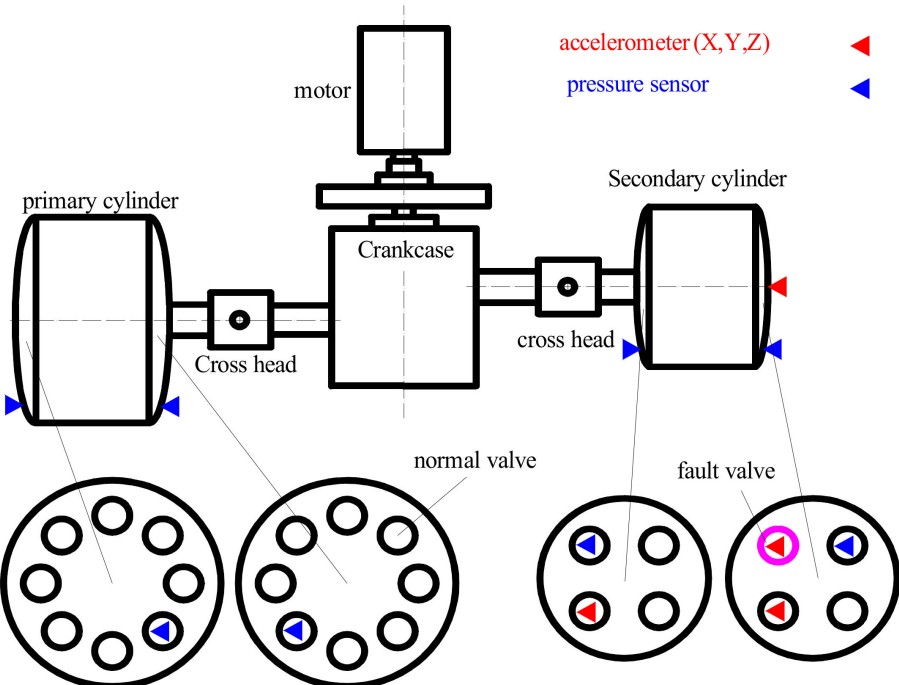

**Figure 9.** No.1 structure and two-stage gas valve measuring point layout.

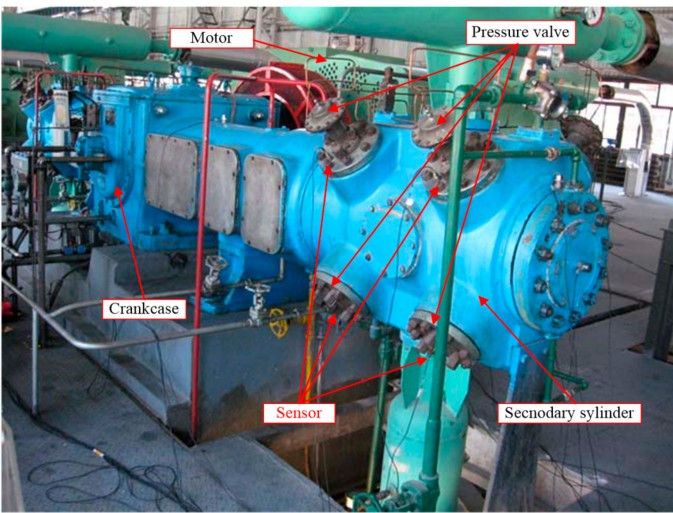

**Figure 10.** Actual positions of the valve and sensors.

During the experiment, the outlet pressure of the secondary gas was 1040 kPa and the inlet pressure was 310 kPa. Besides, the outlet temperature was 104 °C and the inlet temperature was 32 °C and the gas flow was 3611 m$^3$/h. The selection model of the acquisition system was as follows: INV306U-6660 intelligent data acquisition as well as processing analyzer and the INV-1021 program-controlled multi-function signal conditioner of noise (China Orient Institute of Noise & Vibration). The data storage format was stored in an 8-channel, including acceleration (3-channel), pressure (4-channel), and key phase (1-channel). By the way, the key phase signal was the signal received by the sensor installed nearby the spindle as shown in Figure 11. The sensor received a pulse signal when the sensor faced the keyway every time the spindle rotated and the signal is given in Figure 12. When a single-measuring point was used to collect signals, the collected single-channel signal was a vibration signal, which was collected by an acceleration sensor. When multiple measuring points were applied to collect signals, the collected eight-channel signals were composed of vibration signals (three acceleration sensors), pressure signals (four pressure sensors), and key phase signals (one key phase sensor). Furthermore, the data were saved in txt format and each column was a channel data. Under normal conditions, 120,000 data points were collected for each channel and 80,000 data points were collected for each of the three valve states. Figure 13 depicts the acceleration waveform of the vibration signal of channel 1 when the valve disc is broken.

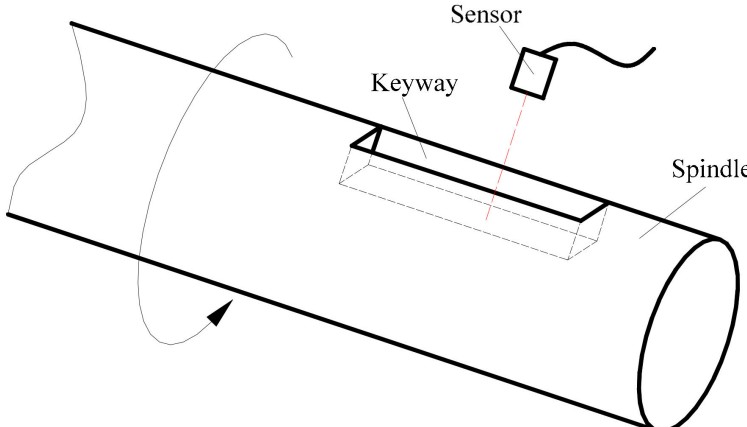

**Figure 11.** Key phase signal acquisition.

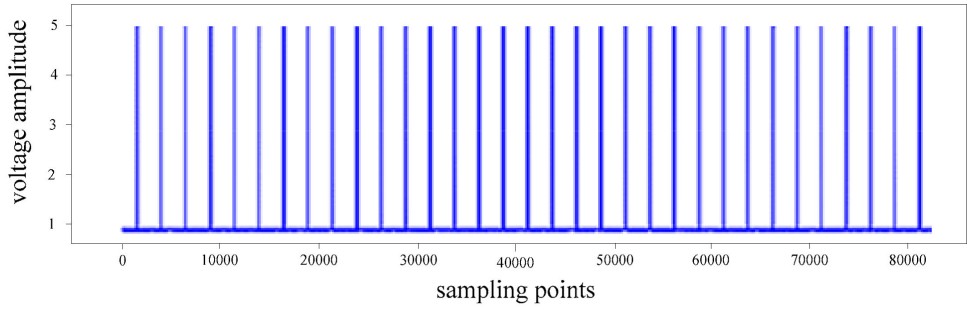

**Figure 12.** Key phase pulse signal.

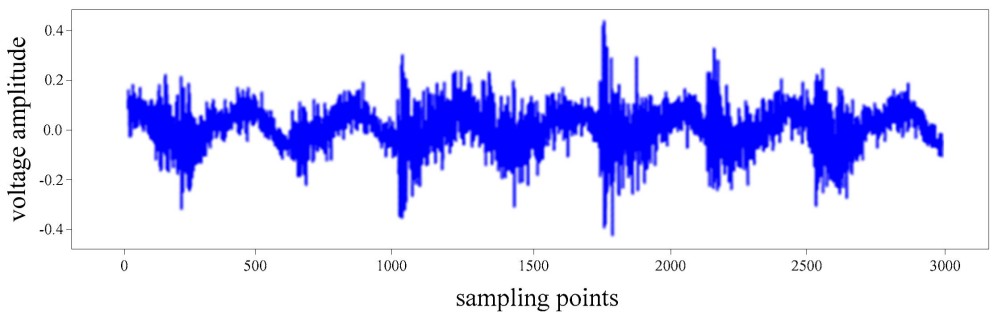

**Figure 13.** Acceleration waveform of the vibration signal.

*6.2. Comparisons of the 1-D CNN Model and Other Typical Methods*

In order to verify the performance of the 1-D CNN single-measuring point model of reciprocating compressor gas valve fault diagnosis, the test set was trained to optimize the model parameters. Next, the 1-D CNN single-measuring and three other models were applied to compare their accuracy, as shown in Table 4.

**Table 4.** 2-D CNN multi-measurement point reciprocating compressor gas valve fault diagnosis model structure.

|  | Power Spectrum Energy (SVM) | Power Spectrum Energy (BP) | Wavelet Packet Energy (BP) | 1D CNN Single-Point |
|---|---|---|---|---|
| Accuracy | 69.08% | 60.53% | 93.75% | 100% |
| Error task | 47 | 60 | 10 | 0 |

Three classic models were used for comparison, including the SVM model with power spectrum energy, BP model with power spectrum energy, and wavelet packet energy. Next, 78 working samples with 4 states (normal valve disc, broken valve disc, broken spring, notched valve disc) were randomly selected and each sample length was 1024 data points as input for the above model. Furthermore, 40 samples of each working condition were randomly selected to constitute the fault training set (160 in total) for model training and samples (152 in total) to constitute the test set of the model.

Table 4 compares the diagnostic effects of the four models on the test set. As can be seen, the 1-D CNN single-measuring point model has the highest recognition rate with an accuracy of 100%, which can effectively diagnose the reciprocating compressor valve failure.

*6.3. 1-D CNN Comparisons of Single-Point and Multi-Point Model*

The test set was used to test the multi-point and single-point models of 1-D CNN under different noise intensities and the comparison results are shown in Table 4. It can be seen from Table 5 that the accuracy of the 1-D CNN multi-point model on the test set was reduced, especially when the SNR was reduced to below 20 dB, and the model accuracy was significantly reduced.

**Table 5.** Comparison of the diagnostic effects of different models on the test set.

| SNR | Failure Recognition Rate and Error Task | | |
| --- | --- | --- | --- |
| | 1-D CNN Multi-Point Model | 1-D CNN Single-Point Model | 2-D CNN Multi-Point Model |
| 5 dB | 47.32 %/1101 | 58.75%/881 | 49.67%/1052 |
| 10 dB | 55.59%/928 | 85.58%/308 | 99.09%/19 |
| 20 dB | 99.43%/12 | 100.00%/2 | 99.19%/17 |
| 50 dB | 99.52%/10 | 100.00%/1 | 99.19%/17 |
| No noise | 99.43%/12 | 100.00%/2 | 99.19%/17 |

The test sets were extracted with different SNR at the output of the dense layer 1 and principal component analysis (PCA) was performed on them to obtain Figure 14. It can be seen that when the SNR is 5 dB, the output of the valve rupture and spring damage faults at the fully connected layer 1 are too close. Consequently, they overlap widely after dimensionality reduction. In addition, the failure data of the spring damage also overlaps with the gap data of the valve plate to a small extent, resulting in a decrease in the accuracy of the model. More spring damage fault data were collected to better extract the characteristics of this type of fault, which can improve model accuracy. When the SNR was 10 db, only a small number of samples were close to the output of the dense layer 1, but the model accuracy was still only 55.59%. Thus, it is judged that the output layer may not be well classified.

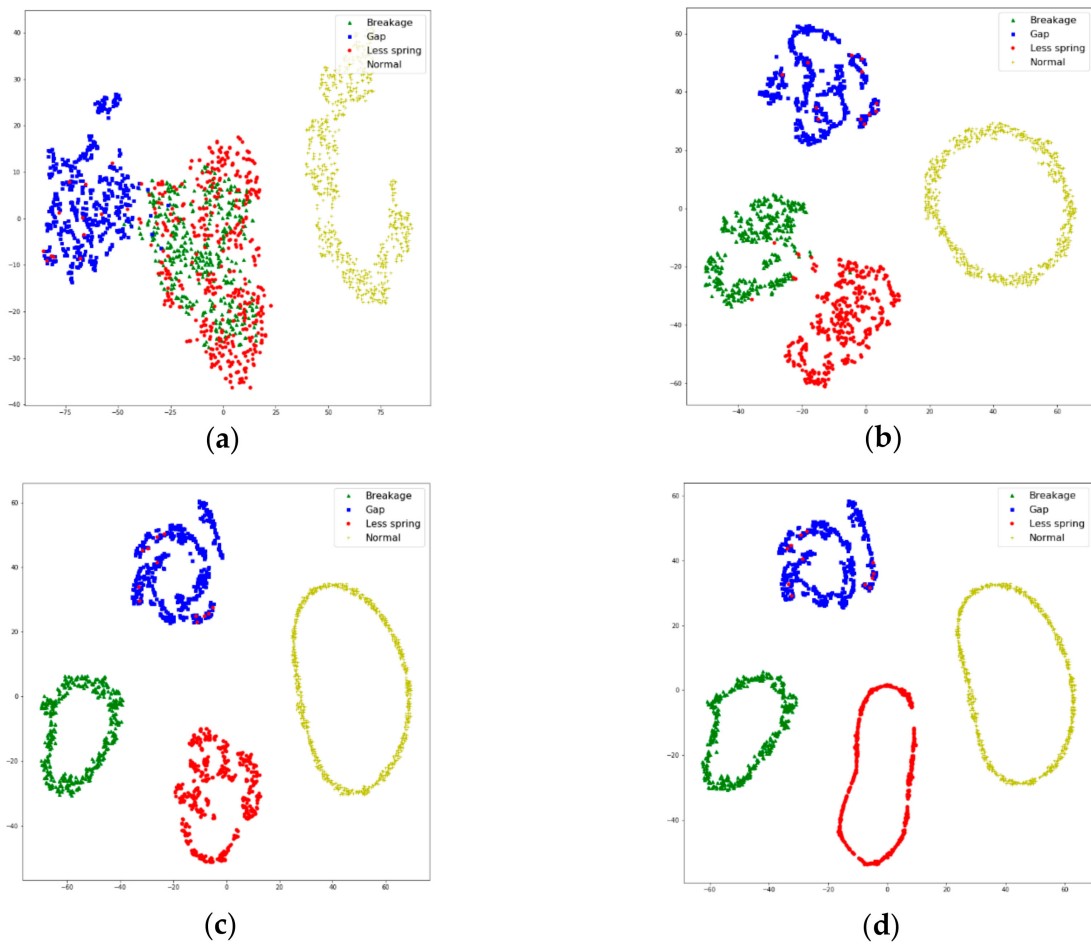

**Figure 14.** PCA of the output of the dense 1 of the 1-D CNN multi-point model of the test set samples under different noise strengths. (**a**) SNR is 5 dB; (**b**) SNR is 10 dB; (**c**) SNR is 20 dB; (**d**) SNR is 50 dB.

### 6.4. 1-D CNN and 2-D CNN Comparisons

The 2-D CNN model proposed in this section was compared with the first two models. The test set was used to test under different SNR and the comparison results are shown in Table 5 above. As can be seen, the performance of the 2-D CNN multi-measurement point model proposed in this section was significantly improved in the 10 dB SNR over the 1-D CNN multi-measuring point model. However, when the SNR was 5 dB, the accuracy of the 2-D CNN multi-measuring point model was only 49.67%, which is lower than the accuracy of the 1-D CNN single-measurement point model. However, when the SNR was 10 dB, the accuracy was significantly improved compared to the 1-D CNN single-measuring point model, and when the SNR was higher, the two types of models behaved similarly. Therefore, the greater the number of fusion points, the accuracy of the model is not necessarily higher. The next chapter will study the influence of the number of measurement points and the selection of the measuring points on the accuracy of the model.

The test set data with different SNR were extracted at the output of the dense layer 1 and PCA was applied to obtain Figure 15. Though the model accuracy was not high when the SNR was 5 dB, the output of the noise-added signal of the dense layer 1 was separable in the high-dimensional space. This result indicates that the model can effectively extract the features of the signal. Therefore, adding fully connected layers can improve the accuracy of the model. From Figure 15, there are a small number of spring damage-type samples the valve disc notch type samples' output at the fully connected layer 1 is too close, which is also the main reason for the small number of model identification errors.

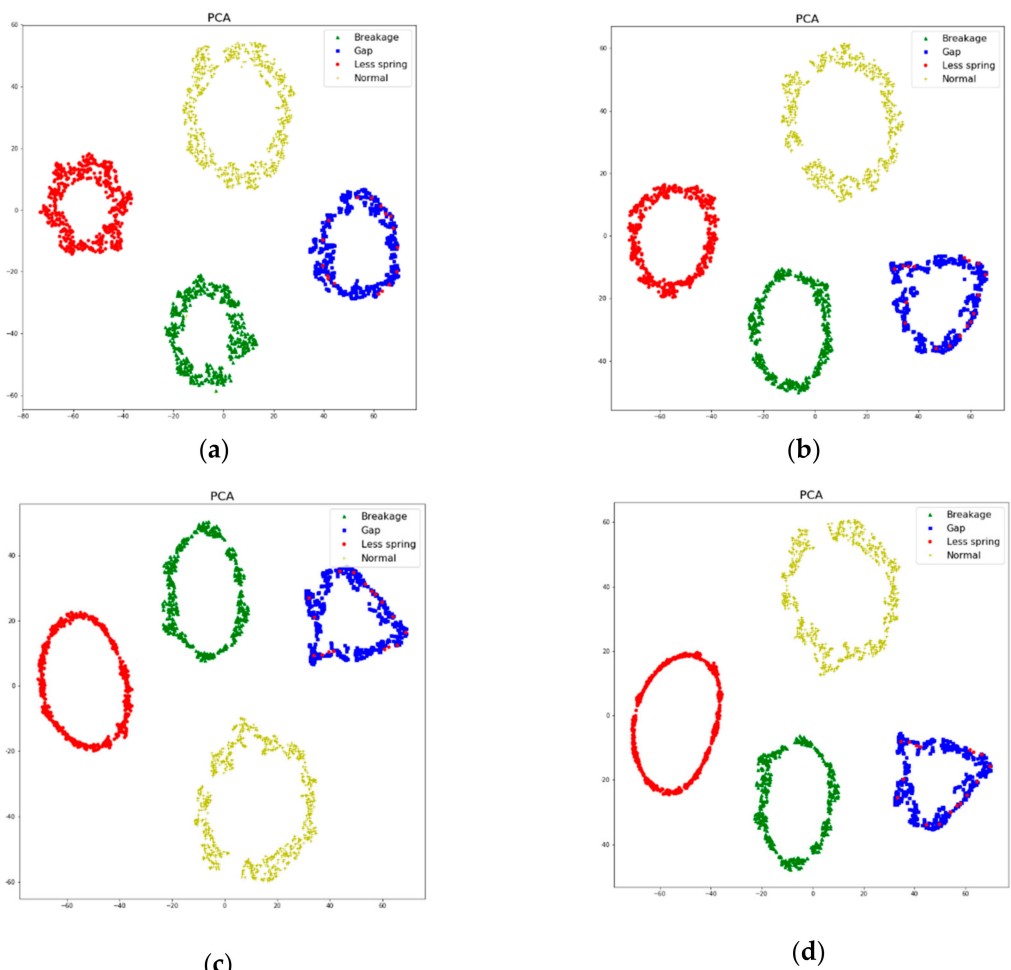

**Figure 15.** PCA of the output of the test set samples in the dense 1 of the 2-D CNN multi-measurement point model under different noise strengths (**a**) SNR is 5 dB; (**b**) SNR is 10 dB; (**c**) SNR is 20 dB; (**d**)SNR is 50 dB.

*6.5. Influence of Fusion of Different Measuring Points*

From the previous section, when the number of measuring points (sensors) increases, the diagnostic accuracy is not necessarily improved. In addition, considering the calculation complexity, the larger number of measuring points means the increase of the model parameters, which will increase the calculation time. Therefore, it is necessary to study how many measuring points can make the calculation time effectively shorten under the condition of ensuring high recognition accuracy in the next section. Further, under the condition that the number of measuring points is fixed, our research can choose which measuring points can have better recognition accuracy.

6.5.1. Influence of the Number of Different Measuring Points

The influence of the number of different test points was analyzed by comparing the accuracy of the model when the number of measuring points was 4, 5, 6, 7, and 8 respectively, corresponding to the above five kinds of training data and the deleted channels, which are shown in Table 6.

**Table 6.** Number of training parameters and training time for different measuring points.

| Number of Measuring Points | 4 | 5 | 6 | 7 | 8 |
|---|---|---|---|---|---|
| Deleted Channel | 3,5,7,8 | 3,6,8 | 3,8 | 8 | Null |
| Training Parameters | 396,796 | 492,028 | 587,260 | 682,492 | 777,724 |
| Training Time | 139.8 | 170.3 | 203.3 | 246.8 | 303.4 |

As the number of channels decreases, the parameters that need to be trained in the model also decrease. Thus, the amount of calculation for each layer also decreases and the length of training time and testing time decreases. The number of training parameters required for the five models and the average duration required for each training set are shown in Table 6. It can be seen from Table 6 that the reduction of the number of measuring points will significantly reduce the model training time. Among them, when the number of measurement points is 4, training time is only 139.8 s, less than half of the training time when the number of measurement points is 8. Therefore, selecting the appropriate measuring points as the input of the model can speed up the model training.

The test results of the five models on the test set are shown in Figure 16. When the number of measuring points is 7, the model has the highest diagnostic accuracy and is superior to the 1-D CNN single-measuring point fault diagnosis model in the previous section. This shows that the key signal of the channel 8 is not suitable for adding the fusion signal as the input of the model. However, as the number of measurement points continues to decrease, the diagnostic accuracy of the model also decreases. Hence, this phenomenon indicates that the data of other channels contains useful feature information. Thus, if these measurement points are deleted, it will reduce the accuracy of fault diagnosis.

6.5.2. Influence of the Fusion of Different Measuring Points

In this section, the focus was on estimating the influence of different measurement points. From the above subsection, it can be known that when the number of measurement points is 7, the model has the best diagnostic accuracy. Therefore, the number of measuring points was fixed at 7 and channel 8, channel 6, channel 4, channel 3, and channel 1 measuring points were deleted to respectively train and test. Next, the accuracy of these cases was compared and the situation with the highest accuracy was found.

As shown in Figure 17, when the key phase signal of the channel 8 was deleted, the model performance was the best. However, when the model with the other channel was deleted, the performance decreased. This phenomenon indicates that only the signal of channels 1 to 7 contains characteristic information handy for fault diagnosis. Combined with the conclusion of the

previous section, it can be known that reasonable selection and deletion of measuring point signals can not only reduce the training time but also improve the accuracy of the experiment.

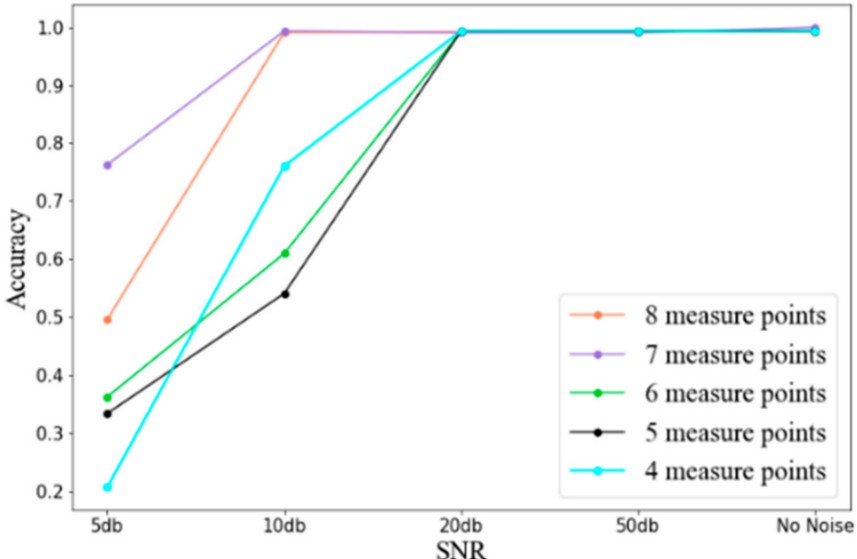

**Figure 16.** Comparison of the diagnosis effect under different numbers of measuring points.

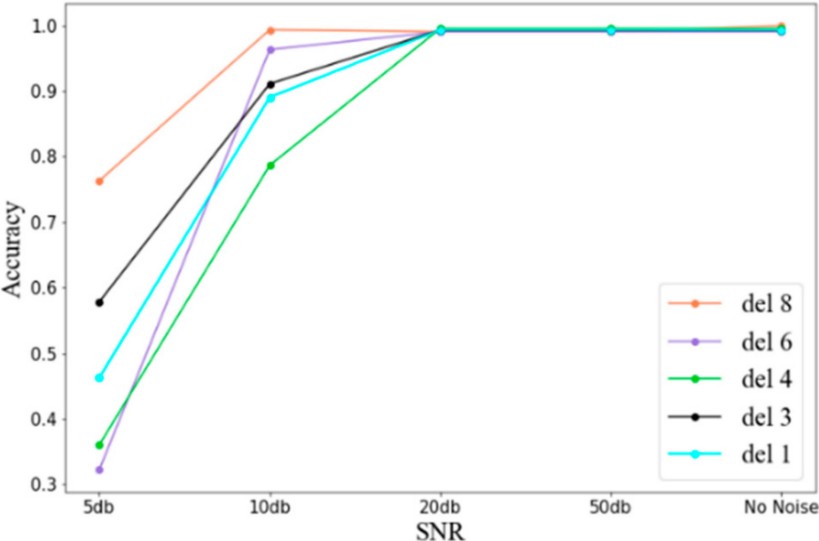

**Figure 17.** Comparison of the fusion diagnosis effect of different measuring points when the number of measuring points is 7.

## 7. Conclusions

In this paper, a CNN-based method was proposed for diagnosing the faults of reciprocating compressors based on single-measuring point vibration signal or multi-measuring points' signal, including vibration, pressure, and key phase signal. In single-point vibration signal, the 1-D CNN model in the deep learning method was used for diagnosis and comparison with three typical methods. The recognition accuracy of 1-D CNN was 100%, which is higher than the other three typical models compared. The results demonstrate the effectiveness of the 1-D CNN model. Next, in order to compare the accuracy of the single-point model and multi-point model, experiments based on 1-D CNN were done to validate it. The experimental results showed that the accuracy of the 1-D CNN single-point

model is higher than the 1-D CNN multi-point model under the 5 SNRs. When the SNR was 10 dB, the difference was the largest and the accuracy of the 1-D CNN multi-point model was only 55.59%. However, the accuracy of the 1-D CNN single-point model was 85.58%. Then, a 2-D CNN multi-point diagnosis model was established and compared with the previous two models. The results showed that the diagnostic accuracy of the 2-D CNN multi-measuring point model was slightly lower than the previous two models when the SNR was higher (20 dB, 50 dB, no noise). However, the accuracy was significantly higher than the previous two models when the SNR ratio was 10dB. When the signal-to-noise ratio was 5 dB, the accuracy of the three models was relatively low, less than 60%. This is enough to prove the effectiveness of the 2-D CNN multi-measuring point model, but the performance of the model needs to be improved. Therefore, the influence of the number of measurement points and the type of measurement points were studied on the diagnosis results under several original signals with different SNRs and it was found that when the number of measurement points was 7 (the key phase signal was deleted), the diagnostic accuracy was the highest(10 dB, 5 dB) in the different measuring points' layout. It was proven that the proper selection and reduction of the measuring point signal can improve the efficiency of diagnosis while ensuring a higher diagnosis accuracy.

**Author Contributions:** Formal analysis, S.X., A.N. and S.L.; Investigation, S.X., M.S. and H.Z.; Funding acquisition, S.L., S.X. and M.S.; Methodology, A.N. and S.X.; Software, A.N. and H.Z.; Supervision, M.S. and H.Z.; Validation, A.N.; Writing—original draft, Z.Z.; Writing—review and editing, S.X. and S.L. All authors have read and agreed to the published version of the manuscript.

**Funding:** This work is supported by the Natural Science Foundation of China (Grant Nos. 51575331), Young teacher special project of Ningde Normal University (Grant No. 2018ZX401, No. 2019ZX401, No. 2019ZX403) and Major Projects of Industry, Education and Research in Universities of Fujian Province (Grant No. 2017H6018). These supports are gratefully acknowledged.

**Conflicts of Interest:** The authors declare no conflict of interest.

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
