# Peer review of "Fault Diagnosis of a Reciprocating Compressor Air Valve Based on Deep Learning"

_applsci, doi:10.3390/app10186596_

Round 1

Reviewer 1 Report

In total, I have only one formal comment on the article. I suggest the authors make a shorter abstract of the paper.

After processing of this comment, I recommend publishing the article

Reviewer 2 Report

In this paper, the use of CNN is proposed to carry out the diagnosis of faults in a reciprocating compressor air valve. The authors carry out different comparisons regarding the use of 1D CNN for single-measurement and multi-measurement and 2D CNN for multi-measurement.

Some remark and questions to the authors:

- The authors do not mention other deep learning methods in the state of the art.

- The authors justify the use of CNN because it is difficult to extract signal features with the traditional feature extraction methods. However, this statement is not entirely correct. Some feature extraction processes, such as statistical feature calculations, are simple to execute and have a low computational cost. The main problem is that when faced with complex scenarios, its characterization capacity decreases, but this does not imply that it is difficult to extract signal features.

- The 2D CNN for single-measurement comparison is missing.

- Please, clarify the meaning of multiple measurement point. Are you referring to signs of vibration, pressure or temperature?

- What does key phase signal mean?

- The process to adapt the 1D CNN for single-measurement to 1D CNN for multi-measurement is not clear enough. Converting input in to signal of 8-channel is not a 2D CNN?

- Clarify the meaning of multi-measuring point. Are they signs of vibration, pressure or temperature?

- In the conclusions the authors mention that the 2D CNN multi-point model has better accuracy than the other models for the SNR scenarios of 10dB and 5dB, but this is not correct.

Reviewer 3 Report

Dear Authors,

The paper present the development of different machine learning algorithms for their application in the air compressor valve. The development of three different algorithms and their verification is interesting:

However, I will suggest some modifications:

1- The references are in general quite old. They should be updated to more recent works.

2- The second paragraph of the introduction should be divided, for example dividing into points the three aspects.

3-The experimental data collection section should be improved with real photos and with an extensive description. The process to obtain the data is a very important step to build the models.

4-The models results should include the error task, not only the accuracy, it could help to the reader.

5-It will be good to add some numbers in the conclusions.

6-Is recommendable to review the English style. Several sentences are quite difficult to understand or have no sense.

Round 2

Reviewer 3 Report

Thanks to all reviewers comments, the paper has been improved, giving a better appearance to the paper and to the work presented.In my opinion, this paper could be published.
